# Specific Septic Complications after Rectal Cancer Surgery: A Critical Multicentre Study

**DOI:** 10.3390/cancers15082340

**Published:** 2023-04-17

**Authors:** Călin Popa, Virgiliu-Mihail Prunoiu, Paul Puia, Diana Schlanger, Mircea-Nicolae Brătucu, Victor Strâmbu, Eugen Brătucu, Hortensia-Alina Moisă, Eduard-Georgian Chiru, Bogdan Vasile Ileanu, Petre Radu

**Affiliations:** 1 Surgery Clinic 3, Regional Institute of Gastroenterology and Hepatology “Prof. Dr. Octavian Fodor”, “Iuliu Hațieganul” University of Medicine and Pharmacy, Croitorilor Street 19, 400394 Cluj-Napoca, Romania; 2Clinic I General and Oncological Surgery, “Prof. Dr. Alexandru Trestioreanu” Oncological Institute, “Carol Davila” University of Medicine and Pharmacy, Fundeni Street 252, 022328 Bucharest, Romania; 3General Surgery Clinic, Clinical Hospital “Dr. Carol Davila”, “Carol Davila” University of Medicine and Pharmacy, Calea Griviței 4, 010731 Bucharest, Romania; 4Center for Health Outcomes and Evaluation, Splaiul Unirii Street 45, 030126 Bucharest, Romania

**Keywords:** colorectal anastomotic abscess and fistula, rectal neoplasm, peritonitis, sepsis, binary logistic model

## Abstract

**Simple Summary:**

Rectal surgery remains burdened by a considerable rate of septic complications. The data of this study show that preoperative radiochemotherapy at the level of the lower and middle rectum allows minimally invasive surgery techniques to be successfully practiced at this level. However, at the same time, this constitutes a contributing factor to postoperative locoregional septic, functional genitourinary and continence complications. This represents the price to be paid for a more conservative and functional surgery. We advocate a personalised treatment that takes both oncological and functional outcomes into account. In this sense, the participation of the patient in the decision-making process is essential, because this makes her/him aware of the possible impact of the treatment on her/his life.

**Abstract:**

The postoperative septic complications in gastrointestinal surgery impact immediate as well as long-term outcomes, which lead to reinterventions and additional costs. The authors presented the experience of three surgery clinics in Romania regarding the specific septic complications occurring in patients operated on for rectal cancer. The study group comprised 2674 patients who underwent surgery over a 5-year period (2017–2021). Neoplasms of the middle and lower rectum (76%) were the majority. There were 85% rectal resections and 15% abdominoperineal excisions of the rectum. In total, 68.54% of patients were operated on laparoscopically, and 31.46% received open surgery. Without taking wound infections into account, 97 (3.67%) patients had abdominal-pelvic septic complications. The aim was to evaluate the causes of the complications. The percentage of suppurations after surgery of the rectum treated by radiochemotherapy was considerably higher than after surgery of the non-radiated upper rectum. The fatality rate was 5.15%. The risk of fistulas was significantly associated with the preoperative treatment, tumour position and type of intervention. Sex, age, TNM stage or grade were not significant at 0.05 the threshold. The risk of fistulas is reduced with low anterior resection, but the gravity of these complications is higher in the lower rectum compared with the superior rectum. Preoperative radiochemotherapy is a contributing factor to septic complications.

## 1. Introduction

Rectal cancer (RC) represents approximately 29% of all tumours located in the colorectal region and has a different behaviour from tumours located in the colon. Thus, for the diagnosis, staging and treatment of RC, the guidelines and treatment protocols are well codified [1]. Rectal dissection with preservation of the mesorectum (Head) and total mesorectal excision (TME) have become the “gold standard” of RC treatment and have had a significant impact on the recurrence of local tumours and the functional results [2]. Surgical resection with negative margins at the histopathological examination is essential to provide the certainty of a correctly performed treatment and to ensure healing [3]. Magnetic resonance imaging (MRI) and trans-rectal ultrasound have substantially improved the staging of RC and allowed us to obtain high-resolution images of the tumour’s extent and the presence and invasion of lymph nodes and organs in the vicinity of the tumour. To these imaging criteria, the presence or absence of vascular invasion and the positive/negative circumferential resection margin (CRM), described on the exeresis pieces and examined histopathologically, have to be added. They, together with the MRI images of the tumour and/or trans-rectal ultrasound, have become important prognostic factors for recurrence and survival [4,5].

In recent years, through a multidisciplinary approach, there have been remarkable advances in the management of RC that incorporate new concepts and technologies and have determined a significant improvement in the biological and functional outcomes of the patients. These efforts have generated substantial progress in local disease control and the survival rate. The introduction of neoadjuvant chemoradiotherapy (CRT) has led to a significant decrease in tumour bulk and the sterilization of lymph node metastases, which provide the grounds for performing sphincter-saving procedures in some cases [4,5].

The surgical treatment of RC for tumours located in the upper and middle parts has been standardised, yet for tumours located in the lower part of the rectum, surgical treatment remains debatable. Lower rectal cancer is defined as a tumour located <6 cm from the anal margin (other studies describe it as <5 cm from the anal margin). For decades, the abdominoperineal excision of the rectum (APER, the Miles procedure) has been the standard of care for the lower RC [6]. The treatment and management of distally located rectal cancer (RC) require a combined effort by the multidisciplinary team (surgeon, oncologist and radiotherapist). The surgical treatment (classic/laparoscopic/robotic) involves total excision of the mesorectum (TME) together with the pelvic lymph node groups and an excision with a circumferential resection margin, which may involve the sphincters and the surrounding organs and which, in spite of all the progress made, may lead to anastomotic complications. Thus, patients are still prone to develop anastomotic fistulas with sepsis, haemorrhage, gas and faecal incontinence, urinary and sexual disorders and pelvic pain postoperatively [4,5,6].

In order to avoid these shortcomings and increase survival, surgeons wanted to develop new techniques and surgical approaches. Thus, nowadays, obese patients who have low rectal tumours and difficult pelvic dissections caused by the narrow operating field and difficult vision can benefit from the TEM (trans-anal endoscopic microsurgery) technique, which consists of approaching the tumour “from the bottom up” [4,5]. The introduction of neoadjuvant chemoradiotherapy has allowed, over time, the development of several types of surgical procedures for saving the sphincter without compromising on oncological principles but with functional results, which are still under discussion. The neoadjuvant treatment can lead to significant tumour regression and a complete histopathological response, allowing the use of conservative organ-preserving treatment such as wait-and-see treatment strategies when there are no residual tumours revealed by the endoscopy and no suspicious lymph nodes or any residual tumours revealed by the MRI. However, with the increase in the importance of the quality of life, a personalised, individualised treatment approach became the rule, which is of the utmost importance for the patient and does not compromise on oncological principles while simultaneously taking the anticipated functional results into account [7,8].

One of the most important specific complications of rectal surgery is pelvic abdominal sepsis, which is rated as having high morbidity (up to 11–12% after post-anastomotic fistulas and 8% after abdominoperineal resection) and a mortality rate of 0.5–1% [5,6]. Sepsis is an immediate postsurgical complication associated with any type of infection that can lead to severe sepsis or septic shock and is an important public health issue (occurring in more than 1% of elective procedures and in more than 4% of nonelective ones). The risk of postoperative sepsis is higher in elderly patients and also in those suffering from diabetes, chronic hepatitis, chronic renal failure and metastatic neoplasms. To decrease its incidence and ensure better postoperative results, one should consider its prophylaxis, the recognition and clinical and paraclinical diagnosis of the complication as quickly as possible, and also the appropriate treatment, which should be started immediately [9].

An anastomotic fistula is initially defined, according to the criteria of Müller, by: faeces externalised on the skin or through the vagina; a fever >38 °C or septicaemia; radiological or endoscopic signs of the fistula; or signs and symptoms of an intraperitoneal abscess or peritonitis [10,11]. There is no consensus regarding the diagnosis of a fistula and its flow based on the size of the anastomotic dehiscence (<1 cm for minor fistulas, >1 cm for major fistulas). The diagnosis is established by clinical examination, biohumoral markers of inflammation, CT examinations, etc. The type of treatment (either conservative or surgical reintervention) is decided according to these parameters. In 2010, the International Study Group of Rectal Cancer (ISREC) recommended a definition and classification system for colorectal anastomotic fistulas. Thus, ISREC defines a fistula following a previous rectal resection as “a defect of the intestinal wall at the site of the anastomosis, which causes a communication between the intra-and extraluminal compartments”. ISREC classified fistulas from A to C based on their management. Thus, there is an important difference in terms of morbidity, mortality, the duration of hospitalization, the cost and therapeutic attitudes among grade A, B and C fistulas. Grade A involves only antibiotic treatment and monitoring; grade B involves, for example, drainage under CT; and grade C requires surgical reintervention [12].

The aim of our study was to evaluate the immediate postoperative outcomes of several types of operations, namely abdominoperineal excision of the rectum (APER) and low or ultralow sphincter-preserving anterior resections (LAR, uLAR) in the treatment of rectal cancer.

## 2. Materials and Methods

Here, the authors present their cumulative experience from three surgical clinics in Romania regarding specific septic complications in patients operated on for rectal cancer. According to the data of the National Institute of Statistics in Romania (Tempo 2021 database, branch POP106A), the two clinical hospitals in Bucharest cover 2 million inhabitants of the city and 2.8 million in the entire South-Muntenia Region. The same source publishes a population of 0.7 million inhabitants for Cluj-Napoca County and 2.5 million for the entire region. Despite the fact that we only hire by region, we mention that Bucharest attracts patients from many other counties, so the coverage area can have over 5.0 million inhabitants. The study focused on a group of 2674 patients operated on during a 5-year period (2017–2021). We noted that the diagnostic and therapeutic indication criteria were uniform for the 3 clinics, as were the surgical techniques and tactics. Neoplasms of the middle and lower rectum (76%) were the majority; the rest were cancers located on the upper rectum, 15–18 cm from the anal verge. Neoplasms located at the level of the recto-sigmoid junction and on the upper rectum were included in a single group. In total, 68.54% of patients (1833) were operated on laparoscopically, and 31.46% (841 cases) underwent open surgery. In the entire study group, 3.67% (97 cases) of abdominal-pelvic septic complications were recorded, namely post-anastomotic fistulas and abdominoperineal resection, without considering wound infections, urinary infections, bronchopneumonia, etc.

The options regarding the type of surgical treatment for patients with lower rectal cancer included abdominoperineal excision of the rectum (APER, the Miles procedure) or low anterior resection with sphincter-saving procedures (low or ultralow sphincter-preserving anterior resections, LAR or uLAR). APER was used in low rectal tumours located 5–6 cm from the anal verge. Indications for APER include ultra-low rectal tumours with an inability to achieve a negative distal margin, external sphincter involvement or levator anus invasion. Those patients with poor baseline sphincter function who have rectal cancer are also suitable for abdominoperineal resection [6].

All the patients included in the study were subjected to a digital rectal examination (DRE) and a sigmoidoscopy/colonoscopy with a biopsy. The neoplasm was staged by computed tomography (CT) examination of the chest and abdomen, magnetic resonance imaging (MRI) of the pelvis and/or endoanal ultrasound for preoperative staging. Following the multidisciplinary examination (by the surgeon, oncologist and radiotherapist), the patients were offered the option of a surgical intervention following a consensus decision after the need for neoadjuvant chemoradiotherapy (CRT) was established. Long-course chemoradiotherapy was administered to T3 and T4 rectal cancers. A laparoscopic or classic open surgical procedure was performed according to the principles of TME. The choice of operative procedure (laparoscopic/classic) depended on the surgeon’s experience and the patient’s preference. The choice of APER or LAR was made after a detailed consultation with the patient, an analysis of the preoperative imaging results, the DRE examination regarding the height of the tumour and the distance from the external anal opening, and according to the recommendations of the medical guidelines. All patients underwent R0 resections. From the point of view of the International Classification of Diseases for Oncology, 3rd Edition (ICD-O-3), for rectosigmoid junction cancers, we used notification C19.9, and for rectal cancer, C20.9 (https://www.who.int/standards/classifications/other-classifications/international-classification-of-diseases-for-oncology, 2nd update, ICD-O-3.2, released 2019—accessed on 30 March 2023).

In the current study, we applied descriptive statistics to characterise the available sample. Next, the Chi-square test and more particular tests such as Phi and Fisher’s exact test were applied to test the association between the presence of complications—abscesses (A) and abscesses with fistulas (AF)—and possible covariates such as age, sex, TNM stage, the tumour’s location, the grading score, the surgical technique and the number of ganglions. Furthermore, a binary logistic model was proposed based on the results of the previous analysis. The odd ratios and the corresponding confidence intervals were reported at *p* = 0.95. Nagelkerke’s R-squared and the percentage of correctly classified cases were reported to assess the quality of the model.

Everywhere in this article, the maximum threshold (*p*-value) for rejecting the null hypothesis of the nonexistence of an association is 0.05 (5%). Thus, associations and differences with *p*-values lower than 0.05 were assumed to be statistically significant. The statistical analysis was performed with the help of IBM SPSS v.21.

## 3. Results

In the study group, 85% (2273 patients) of anterior rectal resections (LAR) and 15% (401 patients) of abdominoperineal rectal resections (APER) were performed. In our group, long-course preoperative radiation was used (50.4 Gy in 28 daily fractions to the tumour and pelvic lymph nodes simultaneously with intravenous 5-fluorouracil (5-FU) at 1000 mg m^2^/day for 5 days during the first and fifth weeks of radiotherapy), followed by TME surgery at 6–8 weeks. We did not practice short-course radiation or radiation for neoplasm cases on the upper rectum. The patients benefited from radiotherapy through IMRT-VMAT technology.

The specific surgical complications noted and reported to us were postsurgical abscesses and fistulas after laparoscopic or classic surgical procedures. From the initial sample of 2674 patients, only 97 (3.67%) presented with postoperative complications such as abscesses and fistulas. The results presented in the next paragraphs describe the subsample of 97 patients with complications of abscesses and fistulas. There were 34 (35.05%) abscesses and fistulas in the procedures for the upper rectum, 19 (19.58%) for the middle rectum and 44 (45.36%) for the lower rectum. Five deaths were recorded: one for the upper rectum, one for the middle rectum and three for the lower rectum (two after APER and one after LAR). Thus, the mortality rate due to specific post-anastomotic complications was 5.15%. For the entire batch studied, it was 0.18%. The percentage of specific postsurgical complications for APER was 7.48% (30 cases) and 2.94% for LAR (67 cases), and the percentage of specific complications compared with the total number of surgical interventions was 3.62% (97 cases out of 2674 patients).

Table 1 lists the specific complications compared with the APER and LAR surgical procedures, performed either laparoscopically or classically, correlated with the rectal segments. Thus, at the level of the upper rectum, 34 surgical interventions (LAR) developed complications. Of these, we noted that 22 were performed laparoscopically and 12 classically. For these complications, conservative treatment was practiced in 28 patients (28.86%) (drainage under CT, lavage), and reinterventions were applied in six patients (6.18%) (laparotomy, drainage, lavage, abolition of the anastomosis, closure of the distal abutment, terminal colostomy). With regard to the neoplasms of the middle rectum, 19 surgical procedures (LAR) developed complications. Two were performed laparoscopically, and 17 were performed via the classic open route. The conservative treatment of complications was applied to 12 (12.37%) patients, and reinterventions were applied to seven (36.84%) patients. The surgical procedures performed at the level of the lower rectum (Figure 1) developed 44 specific complications (30 APER and 14 LAR). Thirteen operations were performed laparoscopically and 31 classically. The 30 cases (30.92%) after APER were only pelvic abscesses, and the 14 (14.43%) after LAR were postoperative abscesses and fistulas. Conservative treatment (drainage under CT, lavage) was performed in 28 patients (28.86%), and reintervention was performed in 16 patients (16.49%), namely laparotomy, drainage, lavage, abolition of the anastomosis and closure of the distal abutment, terminal colostomy, recto-vaginal fistula treatment, vaginal wall suture, etc.

The results shown in Table 2 revealed that age, TNM stage and grading did not influence the probability of abscesses and fistulas (AF) versus abscesses only (A). In each association test, the type 1 error was much larger than the 0.05 threshold. However, some factors seemed to be associated with the difference between abscesses (A) and abscesses + fistulas (AF). Thus, among patients with complications, AF appeared significantly more in the middle and superior locations than in the inferior parts (*p* < 0.001). Furthermore, according to a broad analysis, for the laparoscopic technique, it seems that AF has a greater probability of occurrence than for classic surgery. This can be explained by the large number of laparoscopic interventions: 1833 (68.54%) from the group of patients in this study (2674). Patients with complications and no preoperative RT (radiotherapy) + chemotherapy but who received chemotherapy (rectosigmoid junction and superior rectum) had a lower probability of having AF compared with the no RT + chemotherapy arm. Furthermore, the LAR technique was positively associated with a higher risk of AF in comparison with the APER approach (*p* < 0.001). There was no statistically significant difference for complications between the two sexes (*p* = 0.999).

The results from the initial statistical analysis drove us to build the binary logistic model, with a focus on the significant covariates and avoiding the associations among them, as in the case of surgery type and tumour location (Table 2).

In Table 3, we present the results of the specified logit model. Overall, the model had a good score for classification (83.0%), surpassing the recommended threshold of 0.8. The null model had only a 62.1% capacity for classification. With satisfactory results, we continued to analyse the presence of covariates. The results highlighted the importance of the tumour’s position. Thus, patients with complications and a tumour in the upper part of the rectum are 17 times more likely to develop fistulas (AF) vs. abscesses (A) compared with patients with a tumour located in the inferior part. Furthermore, patients with a tumour position in the middle rectum were 31 times more likely to develop AF vs. A in comparison with patients with a tumour in the inferior rectum (it must be remembered that most surgical interventions were performed on the upper rectum and rectosigmoid junction). One can see that, despite the huge 95% confidence interval of this OR, the results are still statistically significant (*p* < 0.001). The other covariates could not be assumed to be statistically significant, since the 95% CI contained the value of OR = 1, i.e., equal odds.

We found that suppurations after surgery of the radiochemotherapy-treated rectum (i.e., lower and middle rectal surgeries, 63 cases) had a clearly higher percentage than those recorded after non-radiated upper rectal surgery (34 cases).

Regarding the average age of the patients, this was 64.23 years. From the point of view of pTNM staging, most patients with specific postoperative complications were in an advanced stage of the disease, stage III (44 patients, 45.36%), distributed as follows: upper rectum, 17 cases; middle rectum, 5 cases; and lower rectum, 23 cases. Regarding the tumour grade, stage G2 prevailed (58 cases, 65.9%) with an equal distribution among the three segments. The patients who underwent a surgical reintervention (29 cases) had an average age of 65.58 years, and most were diagnosed with stage III tumours (51.72%). The number of reinterventions was higher in stage III tumours (34.09%) compared with stage II tumours (19.23%). The patients with the highest tumour grading percentage (G2) also had the highest number of reinterventions (52.38%) out of the total number of reinterventions. Five deaths were recorded as a result of specific postoperative complications, all of which occurred after open procedures. The average age of the deceased patients was 75.6 years, the patients were in advanced stages of the disease. Three cases were graded as G3, and four of the five cases presented with stage III according to the pTNM classification (Figure 2). Tumor differentiation is graded as well differentiated G1, moderately differentiated G2, or poorly differentiated G3. The “well-and moderately differentiated” grades correspond to low grades, while “poorly differentiated” corresponds to high grades in the two-tiered grading system (WHO Classification of Tumours Editorial Board. WHO classification of tumours: digestive system tumours. 5th ed., Geneva: World Health Organization, 2019 and the International Classification of Diseases for Oncology—ICD-O).

In the total sample of patients with A and AF surgical complications, almost half were in the third stage of the disease. The Chi-square statistical test of the association between stage and tumour position did not reveal significant differences (*p* = 0.833). As expected, patients in the last TNM stage (IV) represented a small percentage in each group, around 5% (one complication from the total of 97 was removed from the comparison of the stage with the location of the tumour because the staging was incomplete).

Of the total number of patients included in the study, 21% of them did not undergo chemoradiotherapy (CRT). For the cases that did not receive CRT, the 3- and 5-year overall survival rates (OS) were 72.2% and 55.55%, respectively. For patients with CRT and surgical intervention, the 3- and 5-year overall survival (OS) were, respectively, 93.75% and 90.90% (the median survival was 55 months). For patients with metastases, the 5-year overall survival was only 30%. For patients with specific septic complications after surgery, survival at one month was 93.18%. We did not register a significant difference in the 5-year overall survival in patients who had specific septic complications (fistula, abscess); this was 90%.

## 4. Discussion

In the results presented above, it can be seen that the number of patients with specific complications of the lower rectum (44) was higher than the number of those with complications of the upper rectum and the rectosigmoid region (34), representing a percentage of 45.36% of the total number of patients with complications. Moreover, 30.92% of the patients who underwent APER with complications presented, these data, which coincide with those in the literature [6]. Practically, the lower one goes with the rectal resection, the higher the number of specific complications (abscesses, fistulas), as well as the number of deaths (three deaths resulting from surgical procedures performed on the lower rectum, compared with one death for the superior rectum and one for the middle rectum). Postoperative fistulas and abscesses were diagnosed clinically and paraclinically (using laboratory tests and imaging) by using an algorithm proposed by our clinic that uses an association between C-reactive protein (CRP) ± the leukocyte count measured from the peritoneal fluid (when there is peritoneal drainage) and blood leukocytes, to which, depending on the results and the clinical examination, we also added the control CT examination [13]. An aspect worth discussing is related to the flow of the fistula. Nowadays, with most surgeons advocating nondrainage after rectal surgery, assessment of the flow is no longer part of the follow-up protocol for a gastrointestinal fistula. We continued to use external drainage, the flow rate being one of the criteria for the therapeutic indications. All the patients with tumours located in the lower rectum underwent neoadjuvant radiochemotherapy, compared with those with tumours in the upper rectum, who received only adjuvant chemotherapy. Patients who underwent radiochemotherapy had a statistically higher risk of developing AF than those who underwent only chemotherapy (Table 2). The number of specific postsurgical complications after APER was higher (7.48%) compared with 2.94% after LAR, and the percentage of specific complications compared with the total number of surgical procedures was 3.62%, a good percentage compared with similar studies in the literature reporting the rate of fistulas to be up to 11–12% [6]. The statistical data showed that age, sex, pTNM stage and grading (G) did not influence the probability of abscess and fistulas (FA) versus abscesses only (A) (Table 2).

As for reinterventions, they were almost double in number for the lower rectum (16 cases) compared with the other segments of the rectum. We can make a similar approximation for the conservative treatment: 28 cases. These complications require a significant number of postoperative days in the ICU and a greater number of hospitalization days; of course, this all translates into the increased costs of the surgical treatment [12]. The group of 97 patients with specific complications had an average hospitalization of 19.2 days (17.35 days for laparoscopic surgery and 20.35 days for classic surgery). Reinterventions increased the number of hospitalization days to 23.51. For patients with CRT and surgical intervention, 3- and 5-year overall survival (OS) were, respectively, 93.75% and 90.90%. The data of our study show that the patients who presented with fistulas and were reoperated on were in an advanced stage of the disease (stage III), with aggressive neoplasms (grade G2). The average age of the deceased patients was 75.6 years (higher than the average age of the group, which was 64.23 years); these patients were in advanced stages of the disease, namely three with grade G3 and four at stage III; all deaths occurred after open surgeries. APER was performed prior to the three deaths that occurred in patients with lower rectal tumours. The introduction of radiochemotherapy for the treatment of lower rectal tumours allowed LAR to be practiced at this level as well, with good results (2.94% complications vs. 7.48% after APER), as shown by the data of our study. However, APER remains the current practice for well-selected cases: patients with the extent of tumour being at the level of the external anal sphincter and the levator ani muscles, those with a partial response after radiochemotherapy and those whose functional anorectal mechanism is compromised by invasion of the intersphincteric space [5]. Laparoscopic LAR (with low or ultralow sphincter-preserving anterior resections, LAR or uLAR) is especially useful for obese patients with a narrow pelvis and bulky residual tumours, where dissection and anastomosis can be difficult. The use of robotic surgery can compensate for the shortcomings of laparoscopy, having the advantages of camera stability, a three-dimensional stereoscopic image and the degrees of freedom of the joints of the working arms. Even so, it is difficult to perform a resection under conditions of oncological security, and thus the trans-anal TME (taTME) technique was proposed, approaching the tumour from the “bottom-up” (“bottom-up” TME). These minimally invasive surgery techniques are credited with good results from the point of view of oncological radicality and the reduced number of hospitalization days (3.46% on average), with specific septic complications of 4.3–7.7% and a mortality rate of 0.5–0.8% [5,14,15].

The anatomical site of the anastomosis remains the most important and significant risk factor for an anastomotic fistula. The more distally an anastomosis is performed, the greater the risk of fistula, with the percentage from the literature being 0.5–18% for colorectal anastomosis and 5–19% for colonic anastomosis, especially for anastomoses performed 5 cm below the anal orifice [12]. Thus, the risk of sepsis increases as we draw closer to the lower rectum. One should keep in mind that along the colon we pass from the intestinal flora with fermenting germs to the putrefying germs in the rectum (Enterococcus, Pseudomonas or Serratia), hence their aggressiveness and the increased septicity, which are some of the causes of the development of anastomotic fistulas and pelvic-abdominal abscesses [15]. These data draw attention to the type of surgical procedure proposed for the patient: anterior rectal resection with low or even ultralow anastomosis, with or without an upstream protective stoma vs. abdominoperineal resection and laparoscopic/robotic/classic procedures [12]. It is recommended, however, at least for frail and elderly (over 70 years old) male patients with multiple comorbidities who are undergoing radiochemotherapy and low or ultralow anastomosis, that they should be protected by an upstream stoma (ileostomy or colostomy) [12]. In our study, fistulas were diagnosed between postoperative days 3 and 35. Considering our experience, we believe that it is useful to drain the pelvic space because this can provide early indications of the development of a fistula [13], a fact confirmed by other specialised studies [15]. Fistulas developing more than 2 weeks after surgery are difficult to differentiate from pelvic abscesses, which are only possible by means of CT, and the diagnosis of these complications (Dutch leakage, DULK) also requires the association of other inflammatory markers such as CRP, procalcitonin [13,15,16,17,18], etc. The conservative treatment uses broad-spectrum antibiotics (at least two from different classes), to which an antifungal can be added in the case of sepsis. It is considered that abscesses under 3 cm in diameter can be successfully managed with antibiotic therapy. Those over 3 cm must be drained (in cases where we have not used an intraoperative drainage tube), percutaneously under CT if possible. The placement of drainage tubes or trans-anastomotic stents and of clips mounted endoscopically at the level of the fistula has also been recommended [15]. In the case of fistulas with a high flow rate and severe sepsis, an urgent surgical procedure such as laparotomy, the abolition of the anastomosis and the creation of a stoma upstream must be performed; in the case of fistulas with a lower flow rate, one possibility is to opt for the restoration of the anastomosis with a protective stoma upstream, washing and drainage [13,15,16,17,18,19,20].

The rectum is the organ with the greatest septicity. If we add to this its susceptibility to neoplasms and neoadjuvant treatments, considerable alterations in the microvascular territories of the pelvic region may be produced. At the same time, the age of the patients with rectal neoplasms is, on average, over 60 years, so their vascular capacity is often altered by atherosclerosis. Furthermore, the low and ultralow resections remove, along with the mesorectum, two of the three sources of the arterial supply to the rectum (the upper and middle rectal arteries). On top of all this, the blood supply of the rectum depends mainly on the superior rectal artery, the middle one being inconstant [21].

Resections remove the rectum almost entirely, with anastomoses often made in the anal canal. A reservoir of some capacity with the properties of adaptive relaxation is thus removed. Following coloanal anastomosis, the colic segment will come into contact with a sphincter resistance system, which is the anal canal. A functional diastasis is created, at least temporarily, between the colon and the anal canal. Rectal resection removes, besides the mesorectum and the two arterial pedicles, the entire apparatus of the levator ani muscles, including the puborectalis fascicle, which creates the anorectal angle that plays an essential role in continence; the angulations of the rectum disappear, configuring a new anatomy and a new functionality. Moreover, the physiological synergism of the mechanisms of defecation is interrupted by the lack of nerve connections between the colon and the anal canal. In the absence of the rectum, the defecation reflex is disturbed, which leads to reduced capacity and compliance of the neorectum. We should not forget the damage to the superior and inferior hypogastric plexuses and to the genitourinary pelvic nerves that may occur during the dissection. Anastomoses 3–4 cm from the anal opening are responsible for the highest rate of functional, fistular and stenotic complications [22].

We should keep in mind that anastomotic fistulas and pelvic abscesses are just some of the complications. Other complications have also been described in the literature (11–63%) [5], which can occur after a low anastomosis and leave their mark on the patient’s quality of life, all of which are described by the low anterior resection syndrome (LARS) [5], consisting of stool frequency (10–15 stools per day), anal (faecal) incontinence, the Wexner score [23], pain during defecation, etc. Their consequences are frequent toilet visits and a preoccupation with defecation, with an impact on the mental and emotional state, wellbeing, intimacy, daily activities and social relationships of the patients. Future technologies may be able to use colon and anal canal manometry for the non-invasive study of motility and thus choose the most effective therapies for LARS patients. This would be an essential step towards improving the quality of life of these patients [22]. Other complications have also been described: dysuria, sexual dysfunction, wound infections, etc. [15,23]. These complications emphasize the need to carefully select the patient who is going to undergo this type of surgery in order to obtain better oncological and functional results [16]. The use of neoadjuvant radiotherapy and the development of fistulas have a long-term negative impact on the anal physiology and functionality [24], influencing the outcomes and making patients undergoing LAR susceptible to anastomotic strictures through pelvic fibrosis, frequent stools and urinary (bladder) incontinence [12,24,25]. This aspect emphasizes the need for an accurate assessment of the anal function before planning this type of surgery, as well as accounting for the patient’s age and activity. It is also useful to provide explanations to the patient and involve him/her in the decision-making process, describing the proposed type of procedure, the functional aspects and the complications that may arise, thus enabling him/her to choose between LAR with a low anastomosis and APER with a definitive stoma.

## 5. Conclusions

Rectal surgery remains burdened by a considerable rate of septic complications. The risk of fistulas is lower in the inferior part of the rectum, but at this level there is a greater number of abscesses. Furthermore, the gravity of these complications is higher in the lower rectum compared with the superior rectum. APER and LAR are feasible procedures for lower rectal tumours.

The data of this study show that preoperative radiochemotherapy at the level of the lower and middle rectum allows minimally invasive surgery techniques to be successfully practiced at this level, reducing the limit of resection from 2 cm, as was performed in the 1980s, to as low as a 2 mm tumour-free zone; however, at the same time, this constitutes a contributing factor to postoperative locoregional septic, functional genitourinary and continence complications. Furthermore, there is the added risk of the inherent complications of a colostomy or ileostomy. This represents the price to be paid for a more conservative and functional surgery. We advocate a personalised treatment that takes both oncological and functional outcomes into account. In this sense, the participation of the patient in the decision-making process is essential, because this makes her/him aware of the possible impact of the treatment on her/his life.

## Figures and Tables

**Figure 1 cancers-15-02340-f001:**
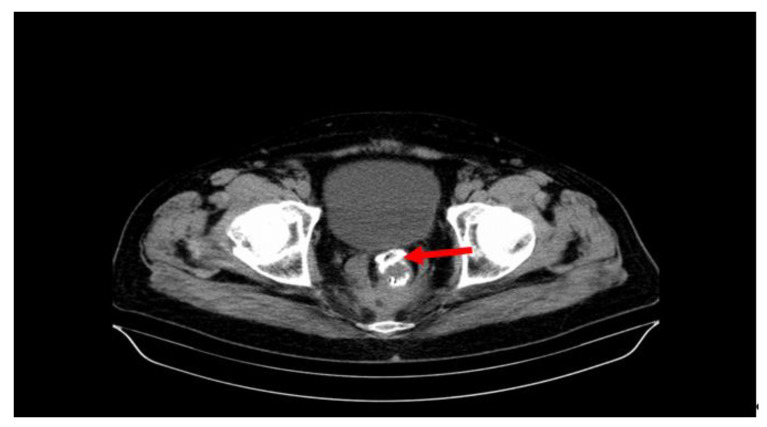
Anterior colorectal anastomotic fistula (red arrow) after laparoscopic low anterior resection (LAR). The image shows the anastomosis clips arranged in a circle.

**Figure 2 cancers-15-02340-f002:**
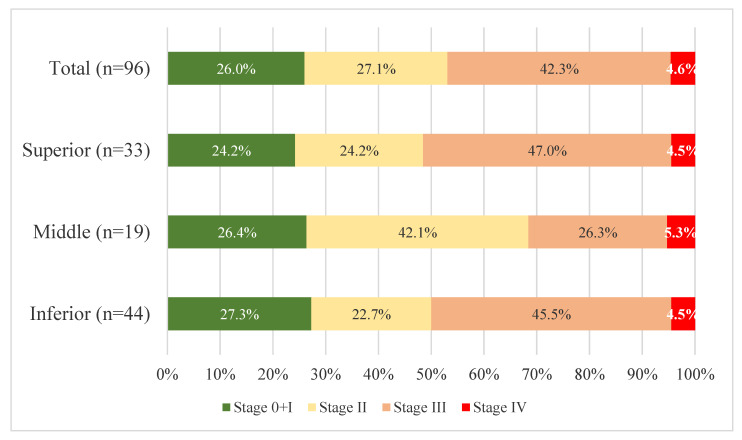
The structure of patients with complications by tumour position and pTNM stage.

**Table 1 cancers-15-02340-t001:** Specific complications related to the type of treatment.

Treatments		Upper Rectum	Middle Rectum	Lower Rectum	Total Complications (97 Cases)
Initial treatment	APER	0 (0%)	0 (0%)	30 (30.92%)	30 (30.92%)
	LAR	34 (35.05%)	19 (19.58%)	14 (14.43%)	67 (69.07%)
	Laparoscopic	22 (22.68%)	2 (2.06%)	13 (13.4%)	37 (38.14%)
	Classic	12 (12.37%)	17 (17.52%)	31 (31.95%)	60 (61.85%)
Treatment of septic complications	Conservative treatment	28 (28.86%)	12 (12.37%)	28 (28.86%)	68 (70.10%)
	Reintervention	6 (6.18%)	7 (7.21%)	16 (16.49%)	29 (29.89%)

APER, abdominoperineal rectal resections; LAR, anterior rectal resections.

**Table 2 cancers-15-02340-t002:** The share of complications by type and sample characteristics.

Category	Covariate/Factor and Reference Category (Ref)	Complications (n = 97)		Association Test Results (Two-Sided *p*-Values)
		Abscess	Abscess + Fistula	
Total		37.1%	62.9%	
Demographics				
Age group	Below life expectancy (<74) (ref)	35.9%	64.1%	*p* = 0.609
	Above life expectancy (≥74)	42.1%	57.9%	
Gender	Females (ref)	37.8%	62.2%	*p* = 0.999
	Males	36.7%	63.3%	
Preoperative characteristics				
TNM class	0 + I (ref)	36.0%	64.0%	*p* = 0.620
	II	30.8%	69.2%	
	III + IV	42.2%	57.8%	
RT and chemotherapy	No (ref)	16.0%	84.0%	*p* = 0.015
	Yes	44.4%	55.6%	
Grading	G1 (ref)	25.0%	75.0%	*p* = 0.367
	G2	37.9%	62.1%	
	G3	50.0%	50.0%	
Tumour position (C19.9; C20.9)	Inferior rectum (ref)	68.2%	31.8%	*p* < 0.001
	Middle rectum	5.3%	94.7%	
	Superior rectum + recto-sigmoid junction	14.7%	85.3%	
Lymph nodes	0 (ref)	34%	66%	*p* = 0.563
	1	45.2%	54.8%	
	2	33.3%	66.7%	
Intervention type and location				
Intervention type	Classic	46.7%	53.3%	*p* = 0.017
	Laparoscopy	21.6%	78.4%	
Location	APER	86.7%	13.3%	*p* < 0.001
	LAR	14.9%	85.1%	
Post interventions	Conservatory	36.4%	63.6%	*p* = 0.826
	Reinterventions	38.7%	61.3%	

RT, radiotherapy; APER, abdominoperineal rectal resections; LAR, anterior rectal resections; and International Classification of Diseases for Oncology (ICD-O-3) for rectosigmoid junction cancers, we used notification C19.9, and for rectal cancer, C20.9.

**Table 3 cancers-15-02340-t003:** Odd ratios of AF vs. A in the sample of patients with complications.

Covariate/Reference Category	Coefficient	OR (95% CI)
Age (continuous)	−0.022	0.978 (0.922–1.037)
Stage II (ref = 0 + I)	−0.170	0.843 (0.167–4.263)
Stage III + IV (ref = 0 + I)	−0.660	0.502 (0.118–2.124)
Superior (ref = inferior)	2.906	18.290 (4.076–82.072) ***
Middle (ref = inferior)	3.452	31.564 (3.217–309.676) ***
Radio and chemotherapy (ref = no treatment)	−0.144	0.866 (0.149–5.036)
Grading G2 (ref = G1)	0.976	2.655 (0.349–20.186)
Grading G3 (ref = G1)	0.331	1.393 (0.255–7.613)
Constant	0.616	1.852
Nagelkerke’s R squared = 0.510; percentage of cases correctly classified = 83.0%		

*** denotes *p*-values < 0.01.

## Data Availability

The authors can confirm that all relevant data are included in the article.

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
