# Peer review of "Specific Septic Complications after Rectal Cancer Surgery: A Critical Multicentre Study"

_cancers, 2023, doi:10.3390/cancers15082340_

Round 1
Reviewer 1 Report (Previous Reviewer 3)
Authors have now responded to my previous criticism and improved the quality of their manuscript. There are some issues of techniques presented in the results section and results in the discussion, but at this point I can approve the scientific content of the manuscript.
Reviewer 2 Report (Previous Reviewer 4)
Introducing these additional data improved the clarity and the content of the manuscipt. It is always interested to read about complications and their impact, therefore I believe we can procede with publication.
This manuscript is a resubmission of an earlier submission. The following is a list of the peer review reports and author responses from that submission.
Round 1
Reviewer 1 Report
1. Abstraction section, the definition of “rectum amputation” is unclear. I don’t think that there is a surgery method of rectum amputation. The authors should use the word precisely.
2. According to the article, fistula formation and abscess is higher and important complications. However, the fistula formation is usually due to rectal cancer metastasis. From the article, one cannot see whether these fistula was cancer involvement or not?
3. From the title, “specific septic complications” after rectal cancer surgery and after reading the whole article, one still cannot get what specific septic complications. Could the authors be more specific?
4. Also, the effects of these complications were unclear. There was study related that “the presence of peritumoral abscesses and fistulas in patients with locally advanced rectal cancer is not associated with increased toxicity or inferior clinical outcomes after preoperative”. PMID: 34825253
5. APR are well known related to more complications to LAR.
6. Also, the conclusion of preoperative radiochemotherapy is a contributing factor to post operative septic complications is fault. These couldn’t be compared due to different position of rectum (upper more movable and lower with peri tissue). The surgery methods are different as well.
Author Response
Please see the attachment
Answer review 1
Good morning.
Thank you for being kind enough to review our article. Your advice is welcome and I will make the requested changes
Acknowledgement: We are very grateful for the valuable comments
Comments and Suggestions for Authors
- Abstraction section, the definition of “rectum amputation” is unclear. I don’t think that there is a surgery method of rectum amputation. The authors should use the word precisely.
Yes. I modified in Abstract 15% "rectum amputations" were performed with "abdominoperineal excision of rectum (APER, the Miles procedure)"
- According to the article, fistula formation and abscess is higher and important complications. However, the fistula formation is usually due to rectal cancer metastasis. From the article, one cannot see whether this fistula was cancer involvement or not?
I would like to specify that all the patients taken by us in the study were diagnosed with rectal cancer. Of these, only 5% also presented metastases (hepatic, pulmonary). Our study focused only on patients who presented complications specific to rectal surgery, namely fistulas and abscesses, without taking into account other complications: wound suppurations, bronchopneumonia, etc.
Yes. Cancer can induce general metabolic and nutritional changes (decrease of total proteins, albumin, etc.), but also local ones (tumor volume, adenopathy, etc.), but, of the large number of patients taken into account, only 3.62% developed complications specific and we wondered what were the causes? The data of our study show that the patients who presented fistulas and were re-operated on were in an advanced stage of the disease (stage III), with aggressive neoplasms (grading G2). The average age of the deceased patients was 75.6 years (higher than the average age of the group of 64.23 years), the patients being in advanced stages of the disease, 3 with grading G3, and 4 with stage III, all occurring after operations open. APER was performed in the 3 deaths that occurred in patients with lower rectal tumors. I specified these aspects in the Discussions chapter
- From the title, “specific septic complications” after rectal cancer surgery and after reading the whole article, one still cannot get what specific septic complications. Could the authors be more specific?
Postoperative sepsis is described as a surgical complication, involving patients immediately after surgical operations, associated with any type of infection that can lead to sepsis, severe sepsis, or septic shock
Yes. We called them "specific complications" because we wanted to refer strictly to complications related to the operative act addressed to the diseased organ (anastomosis of the digestive partners, abdominoperineal resection) and not to complications of the wound, bronchopneumonia, etc. (general complications)
We presented the definition of sepsis in the Introduction, after which we focused on the definition of fistula and abscess after surgical intervention. (I also indicated appropriate bibliographic references). In Material and method, we specified which "specific" septic complications are addressed in our study
- Also, the effects of these complications were unclear. There was study related that “the presence of peritumoral abscesses and fistulas in patients with locally advanced rectal cancer is not associated with increased toxicity or inferior clinical outcomes after preoperative”. PMID: 34825253
The National Cancer Institute (NCI) of the National Institutes of Health (NIH) has published standardized definitions for adverse events (AEs), known as the Common Terminology Criteria for Adverse Events (CTCAE, also called "common toxicity criteria" [CTC]), to describe the severity of organ toxicity for patients receiving cancer therapy. (https://ctep.cancer.gov/protocoldevelopment/electronic_applications/docs/CTCAE_v5_Quick_Reference_8.5x11.pdf (Accessed on March 09, 2018)
We followed in the article the fistulas and abscesses that appeared postoperatively, the consequence of the surgical act addressed to the diseased organ and not the abscesses and fistulas due to a complication of rectal cancer in an advanced state. Article PMID: 34825253 refers to fistulas and abscesses of rectal tumors and the toxicity of the treatment.
The complications of the surgical interventions performed by us consisted of anastomotic fistulas with lower or higher flow, in localized or generalized abscesses (peritonitis) and which required conservative treatment or the application of interventional radiology, interventional endoscopy, or surgical reinterventions adapted to the complication
- APR are well known related to more complications to LAR.
Yes. APR is well known related to more complications to LAR and they are serious.
But there were cases when the position, the volume, the locoregional tumor development, the advanced stage of the disease, the presence of adenopathies imposed this. The 3 University Clinics that treat this pathology in particular applied APR to only 15% of the patients, of course taking into account the above criteria (the surgical approach to be followed was also presented to the patients, and the decision was made only with their consent). The study carried out by us also has a "control" role, namely whether the 3 clinics practice a modern surgery, referring to the data presented in the literature of clinics with a similar pathology, and what is the "price" for a more modern surgery conservative and more functional.
- Also, the conclusion of preoperative radiochemotherapy is a contributing factor to post operative septic complications is fault. These couldn’t be compared due to different position of rectum (upper more movable and lower with peri tissue). The surgery methods are different as well.
Yes. The data of the study showed that preoperative radiochemotherapy at the level of the lower rectum allowed minimally invasive surgery techniques to be successfully practiced at this level, but at the same time it is a favorable factor of postoperative septic complications, compared to the upper rectum where only chemotherapy was practiced . These data are also confirmed by other studies in the literature (bibliographic indices 24, 25 ) 24. Feeney G, Sehgal R, Sheehan M, Hogan A, Regan M., Joyce M, and Kerin M. Neoadjuvant radiotherapy for rectal cancer management. World J Gastroenterol. 2019 Sep 7; 25(33): 4850–4869. two: 10.3748/wjg.v25.i33.4850
- Tabchouri N, Eid Y, Manceau G, Frontali A, Lakkis Z, Salame E, Lecomte T, Chapet S, Calais G, Heyd B, Karoui M, Alves A, Panis Y and Ouaissi M. Neoadjuvant Treatment in Upper Rectal Cancer Does Not Improve Oncologic Outcomes But Increases Postoperative Morbidity. Anticancer Research June 2020, 40 (6) 3579-3587; DOI: https://doi.org/10.21873/anticanres.14348
We advocate a personalized treatment which takes into account both the oncological and functional outcomes. In this sense, the participation of the patient in the decision-making process is essential, because this makes him aware of the possible impact of the treatment on his life.
Yes. I made the necessary corrections to the text in English (number of English editing ID: English-61798)

Reviewer 2 Report
Desi este un studiu retrospectiv , se analizeaza o statistica mare din 3 clinici din Roamania respectand criterii similare de management.
Abordul miniminvaziv este utilizat intr-un numar mare de cazuri.
Prelucrare statistică buna.
Procent mare de rezectii de rect , preocupare pentru o chirurgie functionla.
Complicatii mai multe, dar fara sa impuna reinterventii numeroase.
Fistulele sunt legate de tratamentul neoadjuvant si localizarea tumorii.
Concluziile sunt pertinente.
Bibliografia este actuala.
Author Response
Please see the attachment
Buna ziua.
Va mulțumesc ca ați acceptat sa ne recenzați acest articol.
Într-adevăr este un număr mare de pacienÈ›i luaÅ£i in evidenta de la 3 Clinici Universitare din Romania care abordează patologia respectiva, iar datele prezentate sunt in concordanta cu datele studiate in literatura de specialitate.
ComplicaÅ£iile apărute in urma intervenÅ£iilor chirurgicale reprezintă preÈ›ul de plătit pentru o intervenÈ›ie chirurgicală mai conservatoare È™i funcÈ›ională. SusÈ›inem un tratament personalizat care să È›ină cont atât de rezultatele oncologice, cât È™i de cele funcÈ›ionale. În acest sens, participarea pacientului la procesul decizional este esenÈ›ială, deoarece aceasta îl face conÈ™tient de posibilul impact al tratamentului asupra vieÈ›ii sale.
Va mulțumesc pentru aprecierile făcute
Am tradus si efectuat corecturile necesare in engleza conform of English editing ID: English-61798)

Reviewer 3 Report
The strengths of this paper are:
-A multicentre study from Romania with a large sample n=2674 over 5 years
-Treatment is done with TME surgery and preoperative chemoradiotherapy available
-Paper seems to be designed to detect abscesses with or without fistulae (AF), a specific complication of rectal cancer surgery
However, the are major issues in the presentation of the results, as outlined below. The focus of the paper is also very narrow (AF incidicence) in comparison of the score of the current journal. Therefore I must unfortunately suggest submission to another journal. I hope my comments will help the authors to improved their paper.
-The title refers to septic complications, which include AF, but naturally much more
-There is no description of the three hospitals, on which cities they are located and what are their population coverage. Most importantly, which IRB has approved the current study? Comparison between the three hospitals would also be informative.
-The aim of the study is very narrow (AF) and the paper is too lengthy. Suggest short communication format. Report the time-frame during which AF is collected, as well as overall mortality
-Correct radiotherapy term in chemoradiotherapy or short-course radiotherapy. Please describe whether VMAT/IMRT is used or not.
-The text and abstract is repetitive. Especially results 1st chapter and the entire discussion. Figure 1 is unnecessary. Figure 3 should be a table. Do you refer to pTNM or cTNM?
-Can you provide R0 resection rates? Add to table 2 model. What are "ganglions"? Also provide exact p-values are hazard ratios, instead of "p<0.05".
-Table 3, analyze grading 1-2 versus 3 (high grade)
-Provide general overall survival graphs according to chemoradiotherapy/no and APER/LAR to assess overall results of multimodality treatment
Author Response
Please see the attachment.
Answer review 3
Good morning.
Thank you for being kind enough to review our article. Your advice is welcome. I would be happy if the answers provided will determine the positive appreciation of our article to be included for publication in this prestigious journal
Acknowledgement: We are very grateful for the valuable comments of the anonymous reviewer who took care of this paper
- The title refers to septic complications, which include AF, but naturally much more
Postoperative sepsis is described as a surgical complication, involving patients immediately after surgical operations, associated with any type of infection that can lead to sepsis, severe sepsis, or septic shock
Yes. We called them "specific complications" because we wanted to refer to complications related to the operative act addressed to the diseased organ (anastomosis of the digestive partners, abdominoperineal resection) and not to complications of the wound, bronchopneumonia, etc.
We presented the definition of sepsis in the Introduction, after which we focused on the definition of fistula and abscess after surgical intervention. (I also indicated appropriate bibliographic references). In Material and method, we specified which "specific" septic complications are addressed in our study
- There is no description of the three hospitals, on which cities they are located and what are their population coverage. Most importantly, which IRB has approved the current study? Comparison between the three hospitals would also be informative.
3 University Clinics participated in the study. One from Cluj-Napoca and 2 from Bucharest. All 3 clinics have colorectal cancer as the dominant pathology, have multidisciplinary consultations, surgeon, oncologist and radiotherapy, and have been approved by the ethics committees, all patients being subject to informed consent. The Cluj University Clinic belongs to the Institute of Gastroenterology and Hepatology, the Carol Davila Surgery Clinic has colorectal cancer as its main pathology, and the I Surgery Department of the Bucharest Oncology Institute is dedicated especially to colorectal cancer surgery
¹Surgery Clinic 3, ¹Regional Institute of Gastroenterology and Hepatology „Prof. Dr. Octavian Fodor”, Cluj-Napoca, Romania, ¹"Iuliu HaÈ›ieganul" University of Medicine and Pharmacy, Cluj-Napoca, Romania
²General Surgery Clinic, ²Clinical Hospital "Dr. Carol Davila”, Bucharest, Romania
³Clinic I General and Oncological Surgery
³„Prof. Dr. Alexandru Trestioreanu” Oncological Institute, Bucharest, Romania
²×³³„Carol Davila” University of Medicine and Pharmacy, Bucharest, Romania
“The study was conducted in accordance with the Declaration of Helsinki, and approved by the Institutional Review Board (or Ethics Committee) of NAME OF INSTITUTE: 1. Regional Institute of Gastroenterology and Hepatology “Prof. Dr. Octavian Fodor”, Cluj-Napoca, Romania, nr.219/06.03.2023; 2. Clinical Hospital “Dr. Carol Davila”, Bucharest, Romania; nr. 2684/27.02.2023; 3. “Prof. Dr. Alexandru Trestioreanu” Oncological Institute, Bucharest, Romania, nr 20477/02.09.2022”
According to Romanian National Institute of Statistics data (Tempo database 2021, branch POP106A). The two institutes from Bucharest covers 2 mil. Inh. from the city, and broadly 2.8 mil. inh in the whole South-Muntenia Region. The same source publishes a population of 0.7 mil inh. For Cluj-Napoca County and 2.5 mil. for the entire region. Despite, the fact that we employ only to regions, we mention that Bucharest attracts patient from many other counties, thus the coverage area may be over 5.0 mil. inh.
In the current database.
We conclude that, due to their vast area of coverage, these centres offer a satisfactory variability on the sample for the treated patients.
- The aim of the study is very narrow (AF) and the paper is too lengthy. Suggest short communication format. Report the time-frame during which AF is collected, as well as overall mortality
Yes. We proposed in the article to address especially fistulas and abscesses after surgical interventions at the level of the rectum. We did not want to take into account the complications after interventions on the colon, nor complications such as wound suppurations, brochopneumonias, anastomosis stenoses, sexual dysfunctions or other complications that we just mentioned. We were interested in why certain patients develop postoperative fistulas and abscesses, what could be the causes and what is the "price" paid for a more conservative and functional surgery
The study group comprised 2674 patients who underwent surgery over a 5-year period (January 2017-December 2021). It should be noted that the diagnostic and therapeutic indication criteria were uniform for the 3 clinics, as were the surgical technique and tactics. Thus, the mortality rate due to specific post-anastomotic complications was 5.15% On the whole batch studied, it was 0.18%. I made the correction in the text of the article
- Correct radiotherapy term in chemoradiotherapy or short-course radiotherapy. Please describe whether VMAT/IMRT is used or not.
Yes, I specified in the results text that all patients underwent long-term radiochemotherapy with the appropriate doses “. We specify long-course preoperative radiation was used (50.4 Gy in 28 daily fractions to the tumor and pelvic lymph nodes simultaneously with intravenous 5-fluorouracil (5-FU), 1,000 mg/m2 daily for 5 days during the first and fifth weeks of radiotherapy), followed by TME surgery at 6-8 weeks. We did not practice short-course radiation or radiation of neoplasm cases on the upper rectum.”
Yes, I specified in the results text that all patients underwent long-term radiochemotherapy with the appropriate doses
At the Radiotherapy Clinics in Cluj and Bucharest, cancer treatment is carried out using IMRT-VMAT technology (Oncological Institutes in Romania have this technology). Some of the patients were treated at the Amethyst Radiotherapy Clinics in Cluj and Bucharest. I added in the text that the patients benefited from radiotherapy through IMRT-VMAT technology
- The text and abstract are repetitive. Especially results 1st chapter and the entire discussion. Figure 1 is unnecessary. Figure 3 should be a table. Do you refer to pTNM or cTNM?
In the Discussions I briefly presented data from the Results to compare them with those from the literature and then to emphasize some conclusions.
Ok, we partially agree with your suggestion, then we removed the Figure 1. In the case of figure 3, we think that visual presentation has a better impact. Moreover, we already have two tables. We would like to keep this figure
Yes, being a retrospective study, we thus had access to the anatomopathological pieces of the resection and the classification is pTNM. I also corrected the text
- Can you provide R0 resection rates? Add to table 2 model. What are "ganglions"? Also provide exact p-values are hazard ratios, instead of "p<0.05".
Thank you for your note. We changed the p-values in the text
All patients benefited from R0 resection rates.
The ganglions - lymph nodes - We modified in the table
- Table 3, analyze grading 1-2 versus 3 (high grade)
Thank you for suggestion. We changed the reference category and we performed again the estimations. As expected, we observe positive signs now for grading and minor changes for the values of other factors. The significance has no improvement
- Provide general overall survival graphs according to chemoradiotherapy/no and APER/LAR to assess overall results of multimodality treatment
In the article, we were primarily interested in the evolution of patients after APER/LAR, with fistula complications and abscesses following surgical treatment, occurring 30 days postoperatively. As we said in the article, postoperative chemotherapy was applied as an adjunctive treatment for the upper rectum, and neoadjuvant radiochemotherapy treatment was performed for the middle and lower rectum. We have not discussed in this article the survival of patients after multimodal treatment, with or without post-surgical fistulas and abscesses. This is the subject of an article to be made later
Table 1 The share of complications by type and sample caracteristics
|
Category |
Covariate/factor and reference category(ref) |
Complications (n=97) |
Association tests Results (two sided p-values) |
|
|
Abscess |
Abscess+Fistula |
|||
|
Total |
37.1% |
62.9% |
|
|
|
Demographics |
||||
|
Age-group |
below life expectancy (<74) (ref) |
35.9% |
64.1% |
p=0.609 |
|
above life expectancy (>=74) |
42.1% |
57.9% |
||
|
Gender |
Females (ref) |
37.8% |
62.2% |
p=0.999 |
|
Males |
36.7% |
63.3% |
||
|
Preoperative characteristics |
||||
|
TNM class |
0+I (ref) |
36.0% |
64.0% |
p=0.620 |
|
II |
30.8% |
69.2% |
||
|
III+IV |
42.2% |
57.8% |
||
|
Radiotherapy and chemotherapy |
No (ref) |
16.0% |
84.0% |
p=0.015 |
|
Yes |
44.4% |
55.6% |
||
|
Grading |
G1 (ref) |
25.0% |
75.0% |
p=0.367 |
|
G2 |
37.9% |
62.1% |
||
|
G3 |
50.0% |
50.0% |
||
|
Tumor position |
Rectum Inferior (ref) |
68.2% |
31.8% |
p<0.001 |
|
Rectum Medium |
5.3% |
94.7% |
||
|
Rectum Superior+ recto-sigmoid jonction |
14.7% |
85.3% |
||
|
Lymph-nodes
|
0 (ref) |
34% |
66% |
p=0.563 |
|
1 |
45.2% |
54.8% |
||
|
2 |
33.3% |
66.7% |
||
|
Intervention type and location |
||||
|
Intervention type |
Classic |
46.7% |
53.3% |
p=0.017 |
|
Laparoscopy |
21.6% |
78.4% |
||
|
Location |
APER |
86.7% |
13.3% |
p<0.001 |
|
LAR |
14.9% |
85.1% |
||
|
Post interventions |
Conservatory |
36.4% |
63.6% |
p=0.826 |
The results shown in Table 1 reveals that age, TNM stage and grading does not influence the probability of abscess and fistula (AF) occurrence against only the abscess (A). In each association test the type one error is much more larger than the 0.05 threshold. But, some factors seems to be associated with differences between A and AF. Thus, among patients with complications AF appears significantly more in the medium and superior location than in the inferior side (p<0.001). Furthermore, on a broad analysis, in the laparoscopy technique it seems that AF has a greather probability of occurrence than in the classic surgery. Then, patients with complications and no preoperative RT+chemoterpay show lower propability to have AF compared with RT+chemo arm. The LAR abord is positively associated with a higher risk of AF by comparison with APER approach (p<0.001)
|
Covariate/reference category |
Coefficient |
OR (95% CI) |
|
Age (continuous) |
-0.022 |
0.978 (0.922-1.037) |
|
Stage II (ref=0+I) |
-0.170 |
0.843 (0.167-4.263) |
|
Stage III+IV (ref=0+I) |
-0.690 |
0.502 (0.118-2.124) |
|
Superior (ref=Inferior) |
2.906 |
18.290 (4.076-82.072)*** |
|
Middle (ref=Inferior) |
3.452 |
31.564 (3.217-309.676)*** |
|
Radio and chemotherapy (ref=no treatment) |
-0.144 |
0.866 (0.149-5.036) |
|
Grading G1 (ref=G3) |
0.976 |
2.655 (0.349-20.186) |
|
Grading G2 (ref=G3) |
0.331 |
1.393 (0.255-7.613) |
|
Constant |
0.616 |
1.852 |
|
Nagelkerke R squared=0.510, percentage of cases correctly classified 83.0% |
||
Table 3 Odd ratios to encounter AF vs. A in the sample of patients with complications
The results from the initial statistical analysis drive us to build the binary logistic model, with a focus on the significant covariates and avoiding the association among them, like in the case of surgical type and tumor location (Table 2).
In the table 3 we observe the results of the specified logit model. Overall the model has a good score of classification, 83.0 % overpassing the recommended threshold of 0.8. The null model has only 62.1% capacity of classification. With satisfactory results, we continue to analyze the presence of covariates.
One can underline the importance of tumor position. Thus, patients with complications, having similar other state conditions, and a tumor in superior part of the colon, have 18 time more likely to develop fistula (AF) vs abscess (A) compared with patients with a tumor position in the inferior part. Complementary, patients with tumor position have 31.6 time more chances to develop AF vs A by comparison with patients having the tumor location the inferior rectum. One may see that despite wide 95% confidence interval of this OR, they are still statistically significant (p<0.001). The other covariates cannot be assumed as statistically significant, since the 95% CI contain the value of OR=1, equiodds.
Yes. I made the necessary corrections to the text in English (number of English editing ID: English-61798)

Reviewer 4 Report
Congratulations on a nice series. What I miss is firstly indications for open and laparoscopic at these institutions. Secondly, indications for APER and percentage of protective stomas. Thirdly, indication for deconnection of anastomosis vs ileostomy and drainage in surgical intervention. Finally, timing of reintervantion and relation to outcome in your series.
The description of possible causes of hypovascularisation and defecation issues due to pelvic floor disruption are very well written. That said, when already mentioned it would be nice to see the quantification of these events in your series - in quality of life metrics.
Author Response
Please see the attachment
Answer review 4
Good morning.
Thank you for being kind enough to review our article. Your advice is welcome and I will make the requested changes
Acknowledgement: We are very grateful for the valuable comments
Comments and Suggestions for Authors
Congratulations on a nice series. What I miss is firstly indications for open and laparoscopic at these institutions. Secondly, indications for APER and percentage of protective stomas. Thirdly, indication for deconnection of anastomosis vs ileostomy and drainage in surgical intervention. Finally, timing of reintervantion and relation to outcome in your series.
The description of possible causes of hypovascularisation and defecation issues due to pelvic floor disruption are very well written. That said, when already mentioned it would be nice to see the quantification of these events in your series - in quality-of-life metrics.
- What I miss is firstly indications for open and laparoscopic at these institutions.
Indications for an open or laparoscopic surgery were made on the basis of the clinical examination, the imaging examination (Pelvis MRI/ transrectal ultrasound, but also represented a choice of the surgeon based on his experience, but the patient's wishes were also taken into account, he being - explained the therapeutic possibilities. These aspects are included in the Materials and Methods chapter
- Secondly, indications for APER and percentage of protective stomas.
The surgical treatment of RC for tumors located in the upper and middle part is standardized, yet for tumors located in the lower part of the rectum, surgical treatment remains debatable. Lower rectal cancer is defined as a tumor located <6 cm from the anal margin (other studies describe it as <5 cm from the anal margin). For decades, the abdominoperineal excision of rectum (APER, the Miles procedure) has been the standard of care for lower RC. The treatment management of distally located rectal cancer (RC) requires a combined effort from the multidisciplinary team (surgeon, oncologist, radiotherapist). The options regarding the type of surgical treatment for patients with lower rectal cancer included Abdomino-Perineal Excision of Rectum (APER - Miles procedure) or the Low Anterior Resection with sphincter-saving procedures (low or ultralow sphincter -preserving anterior resections - LAR or uLAR).
The utility of the abdominoperineal resection lies in the ability to remove the low (defined by tumors within 5 cm from the anal verge) tumor, associated lymphoid tissue, and involved structures from within the deep pelvis. Indications for APR include ultra-low rectal tumors with inability to obtain a negative distal margin, involvement of the external sphincter or invasion of the levator ani complex. Those patients with poor baseline sphincter function with rectal cancer are also well-suited for abdominoperineal resection.
Thanks for the observation. I added this to the article
The indication for protective stomas took into account the experience of the clinic and the attending physician. The percentage of protective stomas for the three clinics was 69.73%
- Thirdly, indication for deconnection of anastomosis vs ileostomy and drainage in surgical intervention.
The decision was made depending on the volume of the collection/abscess on CT, the fistula flow assessed on the drain tube and the therapeutic possibilities according to these aspects: conservative treatment, drainage under CT, clip applied endoscopically to the fistula, the intraoperative aspect of the anatomy and of the pelvic collection that determined the disconnection of the anastomosis or the creation of an ileostomy and drainage. In the discussion chapter I exposed these aspects
- Finally, timing of reintervantion and relation to outcome in your series.
The three hospitals are clinics with experience in colorectal surgery. There are studies carried out and published by them in several articles regarding the optimal moment of re-intervention and which take into account the clinical examination, the biohumoral samples (leukocytosis, C-reactive protein, the number of leukocytes from the peritoneal drainage), the abdominal-pelvic CT images, the average time for the reintervention decision being 2-3 days postoperatively. Thus, a correct and timely decision allowed saving the lives of these patients, a fact that was also reflected in the low number of deaths (5) in patients who suffered these complications. (Prunoiu VM, Marincaş AM, Brătucu R, Brătucu E., Ionescu S, Răvaş MM, Ileanu BV. The Value of C Reactive Protein and the Leukocytes in the Peritoneal Fluid in the Predicting Postoperative Digestive Fistulas. Chirurgia (Bucur). 2020 Mar-Apr;115(2):236-245. doi: 10.21614/chirurgia.115.2.236.)
- The description of possible causes of hypovascularisation and defecation issues due to pelvic floor disruption are very well written. That said, when already mentioned it would be nice to see the quantification of these events in your series - in quality-of-life metrics.
Thank you so much for feedback. Yes, we looked to see what could be the causes that can determine these complications. Regarding the quality of life of these patients, it is a topic that we propose to develop in a separate article. Now we wanted to see the postoperative evolution at 30 days.
Yes. I made the necessary corrections to the text in English (number of English editing ID: English-61798)

Round 2
Reviewer 1 Report
The complications of the surgery is high compared with NEJM (PMID 15496622). Also, I don't think it's fair to compare APR with LAR, let alone that there are many surgeons who perform these methods across 3 different centers.
Author Response
Answer for Reviewer 1 Round 2.
Please see the attachment
Good afternoon
First of all, I want to thank you for taking the time to comment on our article. Any suggestion is welcome for a good development and consistency of our article
I made the changes to the text requested by the reviewers and improved the English translation (English editing ID: English-61798, MDPI manuscript ID: cancers-2225002).
Comments and Suggestions for Authors
- The complications of the surgery is high compared with NEJM (PMID 15496622).
The article: PMID: 15496622, DOI: 10.1056/NEJMoa040694, Preoperative versus postoperative chemoradiotherapy for rectal cancer, N Engl J Med. 2004 Oct 21;351(17):1731-40.
In the article developed by us, the authors addressed the specific septic complications after surgical interventions performed at the level of the upper, middle and lower rectum, namely anastomotic leakage and their consequent abscesses. The percentage of these complications compared to the entire study group (2674 patients) was 3.67% (97 patients), a percentage that we consider acceptable and which overlaps with the data communicated by other authors, as we showed in the Discussions chapter.
In the article to which you refer the percentage of anastomotic leakage is to quote "The rate of anastomotic leakage of any grade was 11 percent in the preoperative-treatment group and 12 percent in the post-operative-treatment group" (page 1737 of the article)
- Also, I don't think it's fair to compare APR with LAR, let alone that there are many surgeons who perform these methods across 3 different centers.
Yes. It is correct, but we do not compare LAR with APER in the article. All 3 university clinics address colorectal cancer and comply with national and international surgical guidelines and indications. The choice of the type of surgical intervention was made according to the stage of the disease, the imaging examinations (MRI, transrectal ultrasound), the preoperative clinical assessment and, of course, the patient's wish.
At APER, only well-selected cases were used, under well-established conditions, as I presented in the article, and which respects the indications and attitude of other authors from the international literature. I quote from our article "APER was used in low rectal tumors located 5-6 cm from the anal verge. Indications for APER include ultra-low rectal tumors with inability to achieve a negative distal margin, external sphincter involvement or levator anus invasion. Those patients with poor baseline sphincter function with rectal cancer are also suitable for abdominoperineal resection [6]."
So, I did not compare the two methods. Each of them addressed a certain stage of locoregional development of the patient's disease. APER was used in few cases, but in advanced stages of the disease and where LAR was impossible to achieve. But, unfortunately, after APER, the specific septic complications were few, but serious.

Reviewer 3 Report
The authors have now significantly improved their manuscript based on their replies. However, for some reasons, the text "v2" is mainly the same as earlier, and I am not sure whether all the changes listed in replies are transferred to the paper.
1. ok
2. "According to Romanian National Institute of Statistics data (Tempo database 2021, branch POP106A). The two institutes from Bucharest covers 2 mil. Inh. from the city, and broadly 2.8 mil. inh in the whole South-Muntenia Region. The same source publishes a population of 0.7 mil inh. For Cluj-Napoca County and 2.5 mil. for the entire region. Despite, the fact that we employ only to regions, we mention that Bucharest attracts patient from many other counties, thus the coverage area may be over 5.0 mil. inh."
This should be added to the manuscript
3. ok
4. ok
5. Figure 1, I still find it unnecessary, suggest removal
6. You should state in the text that all patients are resected R0. The reader does not know it unless stated
7. No, you should use grade 1-2 as reference, as suggested by the current ICD-O-3 classification and compared that to grade 3 (ie. high.grade)
8. Overall survival must somehow be described. Please provide median OS in results for those treated with chemoradiotherapy and those without, along 5-y survival rates.
With these adjustments I believe the text will reach the standard of the current Journal.
Author Response
Answer for Reviewer 3 Round 2.
Please see the attachment
Good afternoon.
I want to thank you for taking the time to review the article. All the observations made by you were welcome and significantly improved the scientific content of this article, gaining consistency and accuracy in the presentation of the obtained data. I hope that, by publishing it, we also contribute with our experience to the understanding of the mechanism of the occurrence of anastomotic leakage in these patients, however difficult to deal with this serious disease - rectal cancer
The authors have now significantly improved their manuscript based on their replies. However, for some reasons, the text "v2" is mainly the same as earlier, and I am not sure whether all the changes listed in replies are transferred to the paper.
Yes. I made the changes to the text requested by the reviewers and improved the English translation (English editing ID: English-61798, MDPI manuscript ID: cancers-2225002). Thank you to the reviewer for their time. The article looks much better
- "According to Romanian National Institute of Statistics data (Tempo database 2021, branch POP106A). The two institutes from Bucharest covers 2 mil. Inh. from the city, and broadly 2.8 mil. inh in the whole South-Muntenia Region. The same source publishes a population of 0.7 mil inh. For Cluj-Napoca County and 2.5 mil. for the entire region. Despite, the fact that we employ only to regions, we mention that Bucharest attracts patient from many other counties, thus the coverage area may be over 5.0 mil. inh."
Yes. I have inserted this paragraph in the text (Materials and Methods) which shows that the 3 clinics serve vast populations in the territory of Romania. The clinics also carry out colorectal cancer screening activities and have gastroenterology departments that can detect cases of colorectal cancer.
“Here, the authors present their cumulative experience in three surgical clinics in Romania regarding specific septic complications in patients operated on for rectal cancer. According to the data of the National Institute of Statistics in Romania (Tempo 2021 database, branch POP106A) the two Clinical Hospitals in Bucharest cover 2 million inhabitants of the city and 2.8 million in the entire South-Muntenia Region. The same source publishes a population of 0.7 million inhabitants for Cluj-Napoca County and 2.5 million for the entire region. Despite the fact that we only hire by region, we mention that Bucharest attracts patients from many other counties, so the coverage area can be over 5.0 million inhabitants. The study focused on a group of 2674 patients operated on during a 5-year period (2017-2021). We noted that the diagnostic and therapeutic indication criteria were uniform for the 3 clinics, as were the surgical techniques and tactics …”
- Figure 1, I still find it unnecessary, suggest removal
Yes. I removed Figure 1 from the text and renumbered the figures
- You should state in the text that all patients are resected R0. The reader does not know it unless stated
Yes. I added and specified in the text that all patients underwent R0 resection
- No, you should use grade 1-2 as reference, as suggested by the current ICD-O-3 classification and compared that to grade 3 (ie. high.grade)
Yes. We modified and used the current ICD-O-3 classification and compared it with grade 3 (high grade)
“From the point of view of the International Classification of Diseases for Oncology, 3rd Edition (ICD-O-3) for rectosigmoid junction cancers we used notification C19.9 and for rectal cancer C20.9 (https://www.who.int /standards/classifications/other-classifications/international-classification-of-diseases-for-oncology, 2nd update, ICD-O-3.2, released 2019)”
“Tumor differentiation is graded as well differentiated G1, moderately differentiated G2, or poorly differentiated G3. The “well - and moderately differentiated” grades correspond to low-grade, while “poorly differentiated” corresponds to high-grade of the two-tiered grading system (WHO Classification of Tumours Editorial Board. WHO classification of tumours: digestive system tumours. 5th ed. Geneva: World Health Organization, 2019 and and International Classification of Diseases for Oncology - ICD-O).
- Overall survival must somehow be described. Please provide median OS in results for those treated with chemoradiotherapy and those without, along 5-y survival rates.
Yes. I also entered data on overall survival as follows:
„Of the total number of patients included in the study, 21% of them did not undergo chemoradiotherapy (CRT). For the cases that did not receive CRT, 3- and 5-year overall survival (OS) were 72.2% and 55.55%. For patients with CRT and surgical intervention, 3- and 5-year overall survival (OS) were 93.75% and 90.90% (the median survival was 55 months). For patients with metastases, the 5-year overall survival was only 30%. For patients with specific septic complications after surgery, survival at one month was 93.18%. We did not register a significant difference in the 5-year overall survival in patients who had specific septic complications (fistula, abscess), this being 90%.”
